# Antibacterial and Antibiofilm Properties of Native Australian Plant Endophytes against Wound-Infecting Bacteria

**DOI:** 10.3390/microorganisms12081710

**Published:** 2024-08-19

**Authors:** Meysam Firoozbahr, Enzo A. Palombo, Peter Kingshott, Bita Zaferanloo

**Affiliations:** 1Department of Chemistry and Biotechnology, School of Science, Computing and Engineering Technologies, Swinburne University of Technology, Hawthorn, VIC 3122, Australia; mfiroozbahr@swin.edu.au (M.F.); pkingshott@swin.edu.au (P.K.); 2ARC Training Center for Biofilm Research and Innovation, School of Science, Computing and Engineering Technologies, Swinburne University of Technology, Hawthorn, VIC 3122, Australia; 3ARC Training Centre in Surface Engineering for Advanced Materials (SEAM), School of Engineering, Swinburne University of Technology, Hawthorn, VIC 3122, Australia

**Keywords:** endophytes, antimicrobial, antibiofilm, wound dressings, antibacterial additives, natural products, *Staphylococcus aureus*, MRSA

## Abstract

The wound management field faces significant challenges due to antimicrobial resistance (AMR) and the complexity of chronic wound care. Effective wound treatment requires antimicrobial dressings to prevent bacterial infections. However, the rise of AMR necessitates new antimicrobial agents for wound dressings, particularly for addressing bacterial pathogens like methicillin-resistant *Staphylococcus aureus* (MRSA). Endophytic fungi, known for producing diverse bioactive compounds, represent a promising source of such new agents. This study tested thirty-two endophytic fungi from thirteen distinct Australian native plants for their antibacterial activity against *S. aureus*. Ethyl acetate (EtOAc) extracts from fungal culture filtrates exhibited inhibitory effects against both methicillin-sensitive *S. aureus* ATCC 25923 (MIC = 78.1 µg/mL) and MRSA M180920 (MIC = 78.1 µg/mL). DNA sequence analysis was employed for fungal identification. The most active sample, EL 19 (*Chaetomium globosum*), was selected for further analysis, revealing that its EtOAc extracts reduced *S. aureus* ATCC 25923 biofilm formation by 55% and cell viability by 57% to 68% at 12 × MIC. Furthermore, cytotoxicity studies using the brine shrimp lethality test demonstrated low cytotoxicity up to 6 × MIC (25% mortality rate) with an LC50 value of 639.1 µg/mL. Finally, the most active sample was incorporated into polycaprolactone (PCL) fiber mats via electrospinning, with resultant inhibition of *S. aureus* species. This research underscores the potential of endophytic fungi from Australian plants as sources of substances effective against common wound pathogens. Further exploration of the responsible compounds and their mechanisms could facilitate the development of wound dressings effective against MRSA and innovative biofilm-resistant electrospun fibers, contributing to the global efforts to combat AMR.

## 1. Introduction

The human skin serves as a vital barrier, protecting us against external environmental factors and microbial invasions. When this protective barrier is damaged, it allows microorganisms to penetrate the surface, potentially leading to infections. In severe cases, this can lead to life-threatening complications, such as the loss of essential body fluids and nutrients. This issue emphasizes the global challenge of wound care, particularly in the context of chronic wounds that are not going through the natural healing process [1]. Data have confirmed that more than 305 million people suffer from wounds worldwide and the annual cost of chronic wound care is expected to reach USD 80 billion globally by 2024 [2].

The importance of this issue has sparked growing enthusiasm among researchers to investigate various wound dressing options. These dressings are designed to absorb exudate, combat the infection, relieve the pain, promote autolytic debridement, or even provide and maintain a moist environment at the wound surface. However, no single wound dressing fulfills all of these objectives comprehensively. The selection of wound dressing depends on factors such as the patient’s health status, wound characteristics (type, location, and depth), amount of exudate, adhesion properties, and economic considerations [3].

Hydrogels, foams, dermal patches, films, nanoparticles, hydrocolloids, nanofibers, emulgels, and membranes represent the primary categories of wound dressings. Among them, electrospun fibers and nanofibers stand out as particularly noteworthy types of wound dressings. Their notable attributes, including high surface area, porosity, and drug-loading potential, have motivated extensive research and development efforts [4].

Using an effective wound dressing is expected to enhance the healing process and reduce the likelihood of bacterial infections. The management of bacterial growth and wound infections represents a great challenge in the wound-healing process. In this context, wound dressing materials are often customized with antibacterial agents to reduce these risks during the healing process [5]. However, it is noteworthy that the emergence of AMR has introduced significant obstacles to the use of conventional antimicrobial agents in wound dressings. AMR represents one of the most significant contemporary challenges in global public health. The lack of new antibiotics and antibacterial agents, coupled with the widespread distribution and misuse of existing antibacterial compounds, has led to a surge in antimicrobial resistance, which raises concerns about a potential return to the pre-antibiotic era with infections containing multiple-resistant pathogens affecting ten million lives in the world by 2050 [6,7].

Data confirm the presence of *Staphylococcus aureus* (*S. aureus*) as the most common Gram-positive bacteria on wound surfaces with a share of 75.9% [8]. MRSA is a substantial contributor to healthcare- and community-associated infections on a global scale. In healthcare settings alone, 150,000 patients annually within the European Union are estimated to be impacted by MRSA infections. These infections have given rise to additional in-hospital costs of EUR 380 million for healthcare systems in Europe. Moreover, data on bloodstream infections confirm the presence of up to 50% MRSA among *S. aureus* infections [9].

MRSA exhibits resistance to beta-lactam antibiotics such as penicillin, methicillin, and oxacillin. As some of the important conventional antibiotics do not work against MRSA, treatment is normally furthered using alternatives such as glycopeptides, vancomycin, and teicoplanin. It is worth noting that MRSA has been documented to exhibit resistance against some of these antibiotics. Furthermore, the use of effective antibiotics such as mupirocin and clindamycin is restricted due to the risk of resistance development. Consequently, there is ongoing demand for alternative antibacterial agents for the treatment and prevention of MRSA infections on wound surfaces [10].

In recent years, endophytes as one of the most promising alternatives have gained a lot of attention. These sources, as a recognized source of secondary metabolites, show great potential in various medicinal and agricultural applications [11,12]. However, considering that there are numerous plants and endophytes growing in each ecosystem, screening them all would be impossible. Moreover, there is limited information about some specific plants and endophytes. Australian native plants are no exception in this case. There are a lot of promising Australian plant sources that can produce antibacterial compounds through their secondary metabolites. In order to investigate these groups of plants, a focused approach is needed for plant selection. The rich ethnomedicinal history of some of these plants through thousands of years of utilization among indigenous societies in Australia has created a valuable pool of these promising sources.

Indigenous Australians have historically held a wealth of knowledge about potentially medicinal Australian native plants. Although much of this knowledge was eroded during European colonization, several studies now validate the capacity of these plants to generate bioactive compounds [13]. Some of these plant species are *Eremophila longifolia* (snakevine), which was used to treat colds and bruises by Australian aboriginal communities [14,15], *Callitris preissii* (native pine tree), *Eucalyptus camaldulensis* (Red gum), *Eucalyptus microcarpa* (Grey Box), *Pittosporum phylliraeoides* (native apricot tree) [16], and *Gossypium* species [17].

Numerous research papers have centered on investigating the antibacterial potential of endophytes in wound dressings. A notable example is the work of Firoozbahr et al., who shed light on the capacity of endophytic fungi to act as effective antibacterial agents in wound dressings [3].

Furthermore, in a study conducted by Ingrey et al., different plants from Dharawal country within the Gamay National Park—an Australian Indigenous community—were selected as a source for endophytes. Out of the 48 endophyte isolates examined, 19 tested positive for the presence of polyketide synthase or non-ribosomal peptide synthetase, as identified through PCR using degenerate primers. *Dodonaea* and *Corymbia* harbored the most positive endophytes (seven and six positive endophytes, respectively), followed by *Smilax*, *Leptospermum*, and *Melaleuca* (with four, one, and one positive endophyte(s), respectively). The positive endophytes resulting from *Dodoaea* were identified as *Reticulascus clavatus*, *Sphaeronaemella fragariae*, and *Botryosphaeria agaves*. This identification resulted in *Penicillium limosum*, *Aspergillus keveii*, *Cladosporium iridis*, and *Sphaeronaemella fragariae* for the *Corymbia* plant genus. For the *Smilax* plant genus, *Paraburkholderia nodosa*, *Bacillus paramycoides*, and *Talaromyces flavus* were identified as positive endophytes. Finally, these endophytes were identified as *Penicillium limosum* and *Agrobacterium fabrum* for *Leptospermum* and *Melaleuca*, respectively. Among all of the abovementioned endophytes, *Penicillium limosum* (from *Corymbia* bark), *Aspergillus keveii* (from *Corymbia* branch), *Sphaeronaemella fragariae* (from *Corymbia* branch), *Reticulascus clavatus* (from *Dodonaea* leaf), *Sphaeronaemella fragariae* (from *Dodonaea* root), *Botryosphaeria agaves* (from *Dodonaea* stem), *Penicillium limosum* (from *Leptospermum* branch), *Bacillus paramycoides* (from *Smilax* stem), and *Talaromyces flavus* (from *Smilax* stem) were discovered to have antibacterial activity against *S. aureus* with inhibition zones of 4–6 mm, 4–6 mm, 0–4 mm, 0–4 mm, 0–4 mm, 4–6 mm, 6–8 mm, 0–4 mm, and 0–4 mm, respectively.

The findings indicate that Australian bush medicines encompass a variety of microbial endophytes with notable biosynthetic capabilities. These microbial endophytes offer potential as antibacterial agents, highlighting their promising application in therapeutic contexts [18].

The objective of this study was to investigate 32 endophytic fungi, isolated by our group from the leaves of 13 distinct species of Australian native plants, for their antibacterial and antibiofilm activities against *S. aureus* and MRSA species. Initially, EtOAc and aqueous extracts derived from the fungal culture broths were screened for their inhibitory and bactericidal effects. Next, each active isolate was identified through DNA sequencing, followed by an evaluation of their capacity to inhibit biofilm formation. Finally, after being tested for cytotoxicity, the antibacterial activity of various concentrations of bioactive extracts of the most active sample was incorporated into PCL nanofibers. These nanofibers were subsequently tested against *S. aureus* for their antibacterial activity to determine the presence of antimicrobial components.

## 2. Materials and Methods

### 2.1. Preparation of Fungal Extracts from Australian Native Plants

Leaves of 13 types of plants were obtained from a nursery in Mount Evelyn Victoria. These plants were *Eremophila longifolia* (EL), *Eremophila kalbarricarpet* (EKC), *Eremophila kalgoorlie* (EK) *Eremophila maculata* salmon form (EMSF), *Eremophila calorhabdos* x *denticulate* (EC), *Eremophila glabra* subsp. *carnosa* (EGSC), *Eremophila bignoniifolia* (EB), *Eremophila muelleriana* (EMM), *Eremophila glabra* (EG), *Eremophila nivea* (EN), *Eremophila glabra* x *nivea* (EGXN), *Eremophila racemose* x *maculate* (ERXM), and *Eremophila maculata* silver emu bush (EMSEB).

For endophyte isolation, leaves from the selected plants were employed as the primary source. The protocol outlined by Strobel and Daisy [19] and Zaferanloo et al. [20] was followed. The isolation process commenced with sterilization using 70% ethanol within a laminar flow hood to prevent environmental contamination. Following the removal of outer tissues using a sterile scalpel blade, the leaves were cautiously placed on water agar plates. These plates were then incubated at 25 °C for several days until the emergence of fungal mycelia was observed. To establish distinct cultures, hyphal tips were transferred onto fresh potato dextrose agar (PDA) plates. After an additional incubation period at 25 °C for several days, various subcultures were examined and transferred onto new PDA plates to be confirmed as purified. Over a span of 5–7 days, the growth of endophytic colonies became visible, prompting multiple rounds of subculturing to achieve a state of pure culture.

In the next step, all of the samples went through the solvent extraction process. In this method, after growing mycelia in potato dextrose broth (PDB) (incubation for 14 days at 28 °C), the fermentation broth was separated from the biomass using vacuum filtration through Whatman filter paper No. 1. After discarding the biomass, the filtrate was extracted using EtOAc as an organic solvent at room temperature for two rounds with the final mixture composition of 1:1 EtOAc in the fermentation broth. Each round was performed for 10 min with a rotation of 120 rpm. In the final step, the solvent was evaporated with the help of a rotary evaporator (BUCHI, Essen, Germany) set at 45 °C and 0.22 millibar to reach the extracts with a small amount of solvent. The solutions were then transferred to 1.5 mL microcentrifuge tubes, which were placed in a vacuum concentrator (Christ, Osterode am Harz, Germany) at 40 °C to remove the remaining solvent. The mass of each extract was determined by comparing the pre-process mass of each empty tube with that of the tube containing the extract. Finally, the extracts were dissolved in 1 to 1.5 mL of dimethyl sulfoxide (DMSO) (leading to a concentration of 70–100 mg/mL for each extract) and stored at −20 °C for further study [21].

In a separate batch, the aqueous extracts of each fermentation broth were prepared. Following the separation of the fermentation broth from the biomass, as explained in the previous paragraph, the filtrate was stored at −80 °C and then subjected to freeze-drying using a Telstar freeze-drier (Azbil group, Tokyo, Japan) for 4 days at −20 °C. Following the removal of the aqueous solvent, the mass of the remaining extracts was measured by comparing the mass of the empty tube and the tube containing the extracts. Then, the extracts were dissolved in 5% DMSO and stored at −20 °C for further study. The final concentration of each extract was measured to be between 100 and 120 mg/mL. 

### 2.2. Bacterial Cultures

The extracts were tested against *S. aureus* ATCC25923, obtained from the Department of Chemistry and Biotechnology, Swinburne University of Technology, Melbourne, Australia. They were also tested against bacterial colonies of MRSA M180920 provided by the Microbiology Department, Austin and Repatriation Medical Centre, Victoria, Australia. All bacteria were grown in Mueller–Hinton agar (MHA) plates overnight at 37 °C. Then, Mueller–Hinton broth (MHB) was used to prepare liquid cultures from these plates.

### 2.3. Disc Diffusion Assay

The disc diffusion assay was furthered using the European Committee on Antimicrobial Susceptibility Testing (EUCAST) method [22]. MHA plates were prepared using the manufacturer’s instructions. The media were then autoclaved at 121 °C for 15 min and transferred into sterile 90mm Petri dishes. Bacterial colonies were cultured overnight at 37 °C in MHB and then diluted with fresh MHB to match the turbidity of a 0.5 McFarland standard, equivalent to 1.5 × 10^8^ CFU/mL (colony-forming unit per milliliter) in the absorbance at 600 nm (OD_600_ = 0.08–0.1). In a laminar flow biosafety cabinet, sterilized cotton swabs were dipped into the bacterial solution and swabbed over the surface of the agar plates in three directions to ensure complete and uniform coverage of the solution. In the next step, 6 mm sterilized paper discs were placed on the agar surface using sterile forceps. Then, 10 µL of each sample (EtOAc and aqueous extracts in DMSO) as well as appropriate controls were loaded onto each disc. The plates were incubated at 37 °C for 18–24 h and checked for inhibition zones.

### 2.4. Determination of MIC

The investigation for determining the minimum inhibitory concentration (MIC) was furthered using the broth microdilution method described by the EUCAST [23]. In this test, a starting concentration of 40 mg/mL in DMSO was prepared for each sample. Next, 100 μL of each extract was added to the first well of separate rows in a round-bottomed 96-well microtiter plate, while 50 μL of sterile MHB was added to wells 2–12 of each row. Dilutions were performed serially by transforming 50 μL from well 1 to well 2 in the corresponding row, then 50 μL from well 2 to well 3, and so on until well 12. Finally, 50 μL from the last well was discarded. The bacterial inoculums were prepared by diluting an overnight culture at 37 °C in MHB into fresh MHB to match the turbidity of a 0.5 McFarland standard, equivalent to 1.5 × 10^8^ CFU/mL. The bacterial solution was then diluted to 1:20 by adding 2 mL of the suspensions to 38 mL of sterile distilled water, which resulted in a concentration of approximately 5 × 10^5^ CFU/mL in each well. In the next step, 40 μL of additional media was added to each well, followed by 10 μL of the abovementioned bacterial solution. Thus, the range of final concentrations in the wells was between 20 mg/mL (first well) and 0.0098 mg/mL (last well). After incubation for 24 h at 37 °C, the results were obtained via visual inspection of the lowest dilution with no bacterial growth.

### 2.5. Determination of MBC

After testing the growth assay to assess the inhibition concentration of the samples in a 96-well plate, the MBC test was performed for the most active sample by transferring 20 µL of each well to a nutrient agar plate and incubating it overnight at 37 °C. The bacterial growth after a 24 h time frame suggested the presence of the live bacteria in the solution. Thus, the lowest concentration with no bacterial growth was considered as the MBC.

### 2.6. Extraction of Fungal DNA

Following the bioactivity tests on each sample, the top four active samples were selected for identification. Fungal genomic DNA was extracted from the fungal culture that was previously prepared through growth in PDB. Sanger sequencing was furthered using a Quick-DNA^TM^ Miniprep Kit (Zymo Research, Irvine, CA, USA) using an MPI FastPrep-24 Beater and Eppendorf mini spin plus tubes where necessary. The mixture was placed in a FastPrep-24 homogenizer (MP Biomedicals, Santa Ana, CA, USA) for 120 s at 6.0 m/s for cell disruption. After running this step twice to ensure successful cell lysis, each one of the samples was centrifuged for 10 min at 14,000× *g* to remove cell debris. The rest of the process was conducted using the manufacturer’s instructions.

DNA quality was subsequently validated using a Thermo Scientific NanoDrop One spectrophotometer, measuring within the range of 220 to 350 nm, and recording both DNA concentration (µg/mL) and the absorbance ratio of wavelengths 260/280 (A260/280). Following the confirmation of sample quality, the DNA samples were put through PCR. In 200 µL PCR tubes, 2 µL of extracted DNA sample, 1 µL magnesium chloride (MgCl_2), 1 µL of each primer, ITS1 (5′-TCC GTA GGT GAA CCT GCG G-3′) and ITS4 (5′-TCC TCC GCT TAT TGA TAT GC-3′), and 25 µL Platinum^TM^ SuperFiTM II Green PCR Master Mix (Thermo Fisher Scientific, Waltham, MA, USA). By adding HPLC-grade purified water, the final volume of each tube was adjusted to 50 µL. To amplify the target region, the tubes were placed into a thermocycler using 1 denaturation cycle at 95 °C for 5 min, then 35 cycles of 95 °C for 30 s, followed by 59 °C for 45 s and 72 °C for 60 s. For the final extension, 1 cycle at 72 °C for 7 min was used. To confirm the quality of the amplified bands, the PCR products were analyzed using a 2% (*w*/*v*) agarose gel. After purification of the products using an ISOLATE II PCR and Gel Kit (Bioline, Eveleigh, Australia) using the manufacturer’s instructions, all the DNA samples were prepared using the Australian Genome Research Facility (AGRF, Melbourne, Australia). The samples were then submitted to AGRF, and the resulting sequencing data were analyzed using the Basic Local Alignment Search Tool (BLAST) available on the NCBI GenBank database.

### 2.7. Antibiofilm Activity

The fungal extract exhibiting the highest activity was selected for evaluation of its antibiofilm efficacy against the chosen bacterial strains. The experimental procedure assessed its capability to inhibit initial cell attachment and subsequent biofilm formation using the crystal violet (CV) staining method described by Stepanović et al. [24] and the MTT-based biofilm staining method described by Walencka et al. [25] to measure cell viability through cellular metabolic activity.

To prepare bacterial inocula, each strain was initially cultured overnight at 37 °C in tryptic soy broth with glucose (TSBG) within sterile polystyrene tubes (SPL Life Sciences, Pocheon-si, Republic of Korea). The resultant bacterial cultures were then diluted with fresh TSBG to achieve a 0.5 McFarland standard. Subsequently, this solution was further diluted at a ratio of 1:100 in sterile TSBG. The most active sample was prepared in three concentrations of 12  × MIC, 14  × MIC, and 18  × MIC (adjusted with MIC in TSBG) as a stock solution of 50 × their final desired concentration in DMSO. Then, 4 µL of each extract was added into individual wells of a 96-well polystyrene plate (SPL Life Sciences, Pocheon-si, Republic of Korea), followed by the addition of 196 µL of the diluted bacterial suspension. Proper positive and negative controls were prepared by utilizing 200 µL of bacterial suspension and 200 µL of sterile TSBG, respectively. The positive control represented 100% biofilm formation, while the negative control indicated 0% biofilm formation.

In the CV staining method, following a 24 h incubation period at 37 °C, the culture medium was gently removed through pipetting, and the residual biofilm was subjected to two washes with sterile room temperature phosphate-buffered saline (PBS). Next, the biofilms were fixed via exposure to a 60 °C oven for 1 h. Staining was accomplished by adding 190 µL of 0.02% CV to each well, followed by 10 min incubation at room temperature. After removing excess CV through pipetting, each well was rinsed with fresh PBS at room temperature and further washed under gently running distilled water to eliminate the extra stain. In the final step, 150 µL of 33% acetic acid was added to each well and allowed to remain at room temperature until the stain dissolved. The absorbance of each well was then measured at 595 nm using a microplate reader (POLARstar Omega, BMG Labtech, Ortenberg, Germany). All measurements were conducted in triplicates, and the data were reported as the average of these results. The calculation of biofilm formation inhibition was performed in relation to the controls using the following equation:$$Percentage\;inhibition=100-\left(\frac{{Abs}_{595\;}experimental\;well\;with\;extract}{{Abs}_{595}\;control\;well\;with\;no\;extract}\right)\times 100$$

However, in the MTT staining method, after the biofilm formation period, the microplate wells were emptied and refilled with 150 µL PBS per well. In the next step, 50 µL of 0.3% MTT solution (in PBS) was added to each well and the entire mixture was incubated at 37 °C for 2 h. At the end of the incubation period, the unreduced MTT was extracted and replaced with 150 µL of DMSO and 25 µL of glycine buffer (0.1 mol/L, pH 10.2) to dissolve the stain. Then, the color intensity was determined using a microplate reader (POLARstar Omega, BMG Labtech, Ortenberg, Germany) at 550 nm. Biofilm formation inhibition was calculated using the following equation:$$Percentage\;inhibition=100-\left(\frac{{Abs}_{550\;}experimental\;well\;with\;extract}{{Abs}_{550}\;control\;well\;with\;no\;extract}\right)\times 100$$

### 2.8. Cytotoxicity

To gain a deeper understanding of the effects of the extracts on the entire organism, cytotoxicity was assessed using a test commonly known as the brine shrimp lethality test (BSLT). The BSLT is a versatile bioassay capable of detecting cytotoxicity and the presence of bioactive compounds in an extract.

The brine shrimp eggs, sourced from Aqualabs Australia (Sydney, New South Wales, Australia), were subjected to a BSLT assay based on the method developed by Banti et al. [26], with some modifications. The following steps outline the procedure used for this test.

First, 1 g of brine shrimp cysts was soaked in 500 mL natural fresh water for one hour in a 2 L separating funnel facilitated by an air pump. Then, 17 g of marine salt was dissolved in the 500 mL natural freshwater above. The funnel was kept at room temperature and under continuous illumination for 48 h. After 48 h of hatching, the nauplii released from the eggshells were collected via a micropipette and transferred into a small beaker containing NaCl 0.9%. In the next step, an aliquot (0.5 mL), which contained about 10 to 20 nauplii, was added to each well of a 24-well plate. The EtOAc extracts of EL19 in DMSO were tested at various concentrations, with a range of 12  × to 16 × MIC with three replicates per concentration, in addition to a positive control and a negative control. Each sample was first prepared in 50 × final concentration and then diluted 1:50 with the aliquot to reach the required concentration. The positive control contained 50% DMSO to demonstrate 100% mortality and the negative control contained 2% DMSO to match the concentration of DMSO in each well. The plates were maintained at 25 °C and examined after 24 h using a stereo microscope. Larvae were considered alive if they exhibited internal or external movement during 10 s of observation. The mortality rate of brine shrimp larvae was calculated using the following equation:$$M(\%)=\left(\frac{N_{T}-N_{L}}{N_{T}}\right)\times 100$$
in which M is mortality, $N_{T}\;$ is the total nauplii in each well, and $N_{L}$ is the living nauplii with the tested agent after 24 h.

Finally, the mortality rates of the brine shrimps were plotted against Log10 (Concentration) using GraphPad Prism 10 version 10.0.2. The software was used to determine the LC50 value, the 95% confidence interval, and the $R^{2}$ value of the fitted curve.

### 2.9. Fiber Production

The electrospinning process was conducted using an NS1 Inovenso Electrospinner (Inovenso, Istanbul, Turkey) based on Venugopal et al.’s method [27] with some modifications. PCL, with an average molecular weight of 80,000 g/mole (Sigma-Aldrich, Missouri, USA) was dissolved in a 1:3 methanol/chloroform mixture at room temperature with mechanical agitation using a magnet stirrer in order to achieve a homogenous clear viscous solution with a concentration of 15 wt%. Three different concentrations of the most active EtOAc or aqueous extracts were then dissolved in methanol/chloroform and added to the polymer solution prior to the electrospinning process. The solution was placed in a 1 mL plastic syringe, fitted with a needle (21 G), and was put in the electrospinner. PCL nanofibers were produced via electrospinning using 12 kV in a high-voltage power supplier in a sterile environment. The fibers were collected on a flat surface located 12 cm from the charged needle tip. A syringe pump was used to inject the polymer flow into the system at a rate of 1 mL/h. After spinning, the morphology of the produced fibers was investigated using a Zeiss SUPRA 40 VP field emission Scanning electron microscopy (SEM) at 20 keV. Furthermore, the average fiber diameter of the meshes was calculated by random selection of 10 fibers in each image (9 different images for each set of parameters) using ImageJ software version 1.54i.

### 2.10. Fiber Antibacterial Activity

A certain weight (1 mg) of fiber mats with different concentrations of additives (3.716 mg/mL and 1.89 mg/mL EL 19 extracts in 15 wt% PCL) was tested for its antibacterial activity using the disc diffusion assay discussed previously (Section 2.3). *S. aureus* ATCC25923 was prepared by diluting an overnight culture at 37 °C in MHB into fresh MHB to match the turbidity of a 0.5 McFarland standard, equivalent to 1.5 × $10^{8}$ CFU/mL. Next, 1 mg of each modified and control PCL electrospun fiber mat (fiber mats with no additives as the negative control and fiber mats with the concentration of 53.8 mg/mL of tetracycline as the positive control) was cut using a scalpel blade and placed on the MHA plates in triplicate. Considering the evaporated solvent in the electrospinning process, the mass of additives per mg of fibers for the pre-spinning fiber concentrations of 3.716 mg/mL and 1.89 mg/mL were 309.1 µg and 157 µg, respectively. After incubation for 24 h at 37 °C, the results were obtained by investigating the diameter of the zone of inhibition for each sample.

## 3. Results and Discussion

### 3.1. Antibacterial Activity Screening Using the Disc Diffusion Assay

The antibacterial activities of the EtOAc and aqueous-extracted samples were evaluated using bacterial cultures on agar plates. As previously stated, the selection of bacterial strains was guided by statistical insights from the existing literature. In this context, *S. aureus* was chosen due to its prevalence on wound surfaces. Furthermore, to evaluate the efficiency of the extracts against antimicrobial-resistant bacteria, a strain of MRSA was included in the assessment to be tested against the most promising sample. Notably, a number of the samples exhibited substantial antibacterial activity against the chosen bacteria, indicating their potential as effective antimicrobial agents. The average inhibition zone and activity of each sample against *S. aureus* ATCC25923 and MRSA M180920 can be observed in the following figures (Figure 1 and Figure 2).

Screening tests against *S. aureus* ATCC25923 indicated that the EtOAc-extracted samples obtained from EL19 and EMSF 2-2, showed the highest antibacterial activity with inhibition zones of 11 and 7.2 mm, respectively. Given that EL19 demonstrated the highest activity against the selected bacteria, it was subjected to additional screening against a distinct strain of MRSA. EL19 extracts exhibited significant antimicrobial activity against MRSA M180920 (with an average inhibition zone of 6.5 mm). This outcome underscores its potential as a promising agent in combatting antimicrobial resistance.

After visual inspection, the diameter of the inhibition zone was measured, which can be seen in Table 1 and Table 2.

As stated previously, the most active sample (EL19) was tested against MRSA M180920 for evaluation against antimicrobial-resistant bacteria.

As can be observed from the test results, the EtOAc-extracted samples proved to be more effective in comparison with the aqueous crude extracts, suggesting that the active compounds produced by the endophytic fungi are moderately polar in nature, which makes them more soluble in the organic solvent.

Furthermore, it is noteworthy that the screening process is normally labor and time-intensive and needs to be quantified and confirmed with supplementary experiments such as MIC. The interruption in the inhibition zone does not necessarily mean that the compound was inactive, especially if some of the active compounds are slightly less polar than the others, which diffuse more slowly into the culture medium [28]. There was a similar result for EMSF1-1 against *S. aureus* ATCC25923, which was demonstrated with a growth ring in the middle of the inhibition zone. This observation could potentially be attributed to variations in the polarity of different active compounds. The polarity differences might lead to the formation of a growth ring where activity still extends further. Another reason could be the interaction between some compounds that diffuse across the agar plate, causing inactivity in specific diffusion zones. However, there is a need for isolation of the individual compounds to reach a better understanding of active compounds.

### 3.2. Determination of MIC Values

The MIC of the EtOAc extracts of active samples was measured, which is demonstrated in Table 3 and Table 4. The results indicate that the most active sample against *S. aureus* ATCC 25923 was EtOAc extracts obtained from EL 19 with an MIC level of 78.1 µg/mL.

EtOAc extracts obtained from EL 19 as the most active sample against *S. aureus* were chosen for further testing against MRSA M180920. The results indicated that the MIC level for EtOAc extracts of EL 19 will remain the same against the chosen antimicrobial-resistant *S. aureus*, as can be observed in Table 4, which indicates their efficiency in battling the antimicrobial-resistant bacteria.

These findings provide a deeper insight into the extracts originating from EL 19, showcasing their effectiveness against bacteria in terms of quantity, which aids the conservation of the product and avoids non-targeted specific toxicity. Furthermore, they also provide the opportunity to understand their viability compared with commercial antibacterial agents [29].

Given the previous classification of antibacterial activity by Famuyide et al., as good (MIC < 0.1 mg/mL), moderate (0.1 mg/mL ≤ MIC ≤ 0.625 mg/mL), and weak (MIC > 0.625 mg/mL) [30], the antibacterial activity of EtOAc extracts of EL 19 with an MIC value of 78.1 µg/mL was displayed as good against *S. aureus* ATCC 25923 and MRSA M180920. Moreover, EtOAc extracts of EMSF 2-2 and EMSF 1-2 demonstrated good activity against *S. aureus* ATCC 25923. There are a few studies showing similar results for EtOAc extracts. For example, Phongpaichit et al. isolated *Botryosphaeria* sp. and *Phomopsis* sp. from *Garcinia* species. The EtOAc extracts of these two isolates displayed good antibacterial activity against *S. aureus* and MRSA with MIC values of 32 and 64 µg/mL, respectively [31]. In another study, Yahaya et al. investigated the EtOAc extracts of *Aspergillus fumigatus* from the soil of Bayero University Kano. The results indicated an MIC value of 250 µg/mL against *S. aureus* [32], similar to the case for EMSF EtOAc extracts.

### 3.3. Determination of MBC Values

After observing the turbidity caused by the existing strains of bacteria for each one of the MIC wells, the minimum bactericidal concentration of EL19 extracts was investigated. Table 5 demonstrates the results for bactericidal activity.

As can be seen in the table, while the MIC level for EL 19 EtOAc extracts was the same for the chosen strains of bacteria, the MBC values were found to be different. The MBC values for *S. aureus* ATCC25923 were lower than the MBC values for MRSA M180920.

### 3.4. Identification of Active Sources via ITS Sequencing

DNA sequencing of the ITS region was performed for a better understanding of the source plants. The target region for EL19, EL24, EMSF2-2, and EK3 was successfully amplified and resulted in Table 6.

The identification of the active samples was achieved by comparing the existing sequence data that were available in the GenBank database. The results indicated that both EMSF 1-2 and EMSF 2-2 were identified as *Alternaria* species, with EL 19 identified as *Chaetomium* species (accession no. PP455507.1) and EK 3 identified as *Coniochaeta* species. However, EK 3 exhibited a higher E-value; due to the outcomes of the sequencing, there was lower confidence in the identification of the isolate below the genus level.

*Chaetomium* species have demonstrated significant antimicrobial activity in previous studies reported in the literature. They have been proven to be the source of multiple bioactive compounds including chaetoglobosins, epipolythiodioxopiperazines, azaphilones, xanthones, anthraquinone, chromones, depsidones, terpenoids, and steroids [33]. As an example, in a study conducted by Wang et al., the *Chaetomium globosum* CGMCC 6882 endophytic fungi was evaluated for its potential antibacterial effects against *S. aureus*. The outcomes of this study revealed that Glycyrrhiza polysaccharide (GCP), a polysaccharide produced by the mentioned fungi, exhibited an MIC of 0.67 mg/mL. Furthermore, GCP demonstrated antibacterial properties by disrupting the inner membrane and enhancing cell permeability while having no impact on the cell wall. This suggests its considerable potential for application in the pharmaceutical industry [34].

*Alternaria* species have also demonstrated notable antibacterial activity in previous studies included in the literature. Endophytic fungi within this genus have been frequently isolated as endophytes from various plants. A comprehensive study in this regard is the work conducted by Tsyganenko et al., who investigated the biological activity of *Alternaria* species. The study identified and scrutinized the biological activity of various *Alternaria* species. The results indicated that the secondary metabolites extracted from four different sources of *Alternaria* species exhibited activity against *S. aureus* [35].

In a separate investigation conducted by Al Mousa et al., the antimicrobial potential of the endophytic fungi *Alternaria tenuissima* and *Alternaria alternata*, isolated from *Artemisia judaica* L., was examined. EtOAc extracts from these fungi were subjected to testing against a range of Gram-positive and Gram-negative bacteria to assess their antibacterial activity. The outcomes demonstrated that both *Alternaria* species exhibited an inhibition zone ranging from 25 to 30 mm against *S. aureus* ATCC6538P [36].

Lastly, *Coniochaeta* species have exhibited antibacterial activity against Gram-positive bacteria. For example, the study conducted by Han et al. provides insight into this aspect. In their research, several compounds were isolated and identified from the fungus *Coniochaeta cephalothecoides*. The findings revealed that the compound conipyridoins E displayed the strongest activity against the growth of both *S. aureus* and MRSA, with an MIC value of 0.97 µM [37].

### 3.5. Antibiofilm Characteristics

The antibiofilm activity of the EtOAc extracts of EL 19, as the most active sample, was evaluated against the two strains of *S. aureus*. The focus was on assessing the ability to inhibit the initial attachment of bacterial cells to polystyrene surfaces. Three distinct concentrations of extracts were examined ($\frac{1}{2}$ × MIC, $\frac{1}{4}\;$ × MIC, and $\frac{1}{8}$ × MIC). Given that TSBG was used for the biofilm assays, the extract concentrations were adjusted based on the MIC values obtained using this medium. The CV staining method demonstrated that EL 19 extracts significantly inhibited bacterial cell attachment and, to some extent, reduced biofilm formation at most concentrations tested. For MRSA M180920, reductions in biofilm biomass were observed, ranging from 41% to 55%, when tested at $\frac{1}{2}$ × MIC. At the same concentration, EL 19 extracts demonstrated biofilm inhibition ranging from 47% to 59% in *S*. *aureus* ATCC25923, suggesting slightly higher effectiveness against *S*. *aureus* at this concentration. Similar trends were observed at the concentration of $\frac{1}{4}\;$ × MIC, with higher biofilm inhibition against *S*. *aureus* ATCC25923 (ranging from 44% to 57%) compared to MRSA M180920 (ranging from 29% to 36%). At the concentration of $\frac{1}{8}$ × MIC, the biofilm inhibition in MRSA M180920 and *S*. *aureus* ATCC25923 was comparable, ranging from 8% to 25% and 5% to 28%, respectively. The biofilm inhibition in the CV staining method for each bacterium can be observed in Figure 3.

The MTT staining method for EL 19 EtOAc extracts demonstrated significantly lower cell viability compared to the negative control and, to some extent, reduced biofilm formation at the greatest concentrations tested. For MRSA M180920, reductions in biofilm cell viability were observed from 43% to 47%, when tested at $\frac{1}{2}$ × MIC. At the same concentration, EL 19 extracts caused cell viability reductions ranging from 57% to 68% in *S*. *aureus* ATCC25923, suggesting greater effectiveness against methicillin-sensitive *S*. *aureus* at this concentration. Similar trends were observed at $\frac{1}{4}\;$ × MIC and $\;\frac{1}{8}$ × MIC, with greater cell viability reductions against *S*. *aureus* ATCC25923 (ranging from 52% to 56% for $\frac{1}{4}\;$ × MIC and 34% to 48% for $\frac{1}{8}$ × MIC) compared to MRSA M180920 (ranging from 22% to 28% for $\;\frac{1}{4}\;$ × MIC and 17% to 19% for $\frac{1}{8}$ × MIC). The reduction in cell viability induced by the MTT staining method for each bacteria can be observed in Figure 4.

Biofilms represent aggregates of single or multiple species of bacteria enveloped in a matrix consisting of polysaccharides, proteins, and DNA. This matrix serves as a protective housing that shields the bacteria from various environmental pressures. Beyond the physical barrier provided by the polymeric matrix, bacteria within a biofilm undergo transcriptional changes. This adaptive response allows them to react to perceived stressors and invoke resistance mechanisms, thereby protecting the cells from antibiotics and other antimicrobial threats [38,39]. The formation of biofilms serves as a survival strategy for bacteria to adapt to their living environment, particularly in challenging conditions [40]. Within the protective shelter of a biofilm, microbial cells increase their tolerance and resistance to antibiotics and immune responses, causing challenges for the clinical treatment of biofilm infections. Both clinical and laboratory investigations have highlighted a clear correlation between biofilm infections and medical foreign bodies or indwelling devices [41,42]. Clinical observations and experimental studies have demonstrated that using antibiotics alone is not sufficient for eradicating biofilm formations [42]. Furthermore, antimicrobial resistance reduces the efficiency of previously used antibiotics and intensifies the problem. Consequently, the effective treatment of biofilm infections using currently available antimicrobial agents has become a crucial and urgent goal for clinicians. Given the recognition of *S. aureus* as a prominent wound surface bacteria and a major biofilm-forming pathogen [8,43], in this study, EtOAc extracts of *Chaetomium globosum* as the most active sample were assessed against two strains of *S. aureus.* As observed in the results, both bacterial strains were significantly impacted by the chosen extract, resulting in strong antibiofilm inhibition. Comparing these outcomes with previous studies in the literature, it is noteworthy that the ethyl acetate (EtOAc) extracts of the endophytic fungus *Aspergillus*, for example, demonstrated an 80% reduction in the biofilm formation of *S. aureus* NRRL B-767 but at a concentration of 500 µg/mL [44]. However, the range of concentrations for similar biofilm inhibition in EtOAc extracts of *Chaetomium globosum* was recorded to be around 39 µg/mL. This extract was able to inhibit the biofilm formation of both bacteria, which aligns with findings from a prior study, such as the one conducted by Jalil et al. (2021) [45], who revealed that an EtOAc extract of the endophytic fungus *Lasiodiplodia pseudotheobromae* possessed antibiofilm activity against MRSA ATCC33591 at concentrations > 100 µg/mL but stimulated biofilm production at concentrations between 10 and 60 µg/mL. Thus, it is noteworthy that the reduction in biofilm formation by EtOAc extracts of *Chaetomium globosum* was around 39 µg/mL.

### 3.6. Brine Shrimp Lethality Test

The cytotoxicity of the EL19 EtOAc extract was evaluated using the BSLT. This assay provides a reliable method for testing compounds against a whole organism. The mortality rate of brine shrimp was used to determine the cytotoxic limits of various concentrations. The BSLT results indicated that the EtOAc extracts of EL19 caused mortality rates below 30% at concentrations up to 6 × MIC. No mortalities were observed at $\frac{1}{2}$ × MIC (39 µg/mL); however, the mortality rates for 1 × MIC (78 µg/mL), 2 × MIC (156 µg/mL), 4 × MIC (312 µg/mL), and 6 × MIC (468 µg/mL) were approximately 3%, 12%, 15%, and 25%, respectively. These findings suggest that cytotoxicity becomes significant at concentrations above 6 × MIC. The next concentrations tested were 8 × MIC (624 µg/mL), 12 × MIC (936 µg/mL), and 16 × MIC (1248 µg/mL), resulting in mortality rates of 58%, 100%, and 100%, respectively. Mortality rates of the brine shrimps were plotted against the ${Log}_{10}\left(Concentration\right)$ using GraphPad Prism 10. Using the software, the LC50 value was determined to be 639.1 µg/mL with a 95% confidence interval of 583.3 to 744.2 µg/mL and an $R^{2}$ value of 0.9572 for the nonlinear fitted curve.

These results reflect the high biocompatibility of the EL19 extracts at greater concentrations and a wide therapeutic window. The mortality rate data are shown in Figure 5.

Brine shrimp is a simple invertebrate organism widely used in pharmacological studies, particularly for evaluating plant-based bioactive compounds. For example, in a study by Wanyoike et al., five different Kenyan medicinal plants were assessed for their toxicity and antiplasmodial activity. Organic extracts of leaves and roots from these plants, known for their malaria-curing properties in Kenya, were tested using the BSLT. The results revealed that two of the tested plants, *Albizia gummifera* and *Cyathula cylindrica*, exhibited LC50 values of approximately 274 µg/mL and 153 µg/mL, respectively [46].

### 3.7. PCL-Based Fiber Mat Morphology and Antibacterial Activity

The SEM images of the various meshes is demonstrated in Figure 6. By adding the EtOAc extracts of EL 19 to the pre-electrospinning solution, a smaller fiber diameter is achieved, leading to a higher surface area which is more favourable in electrospinning. The average fiber diameters for PCL with no additives, 1.89 mg/mL EtOAc extracts of EL 19, and 3.716 mg/mL EtOAc extracts of EL 19 were 535.9 nm, 249.385 nm, and 195.131 nm, respectively.

The early qualitative set-up for checking the antimicrobial activity of the produced fibers was performed using disc diffusion tests. The disc diffusion assays for the fibers containing different concentrations of EL19 were tested against *S. aureus* ATCC25923 and resulted in the inhibition zones as shown in Figure 7 and Table 7. The inhibition zones of the tested samples of PCL fibers mixed with EL19 at concentration of 3.716 mg/mL (309.1 µg additive per 1 mg PCL) and 1.89 mg/mL (157 µg additive per 1 mg PCL) were 1.4 ± 0.1 mm and 0.95 ± 0 mm against *S. aureus* ATCC25922, respectively, as well as 1.45 ± 0.05 mm and 0.6 ± 0.05 mm against MRSA M180920, respectively. The results indicated that the fiber mats successfully released their antibacterial additives into the MHA plates with low standard deviations in the zones of inhibition. Moreover, increasing the concentration of the additives increases the average inhibition diameter of the produced fibers.

In the initial phase of assessing the activity of the antibacterial agents within the application area, in vitro studies on the fiber mats were conducted using a disc diffusion assay. These studies not only confirmed the presence of antibacterial additives but also demonstrated successful release into the surrounding bacterial biofilm on the agar plate. Similar investigations have been undertaken to evaluate the antimicrobial activity of additives in PCL fibers. For example, in a study by Adeli-Sardou et al. [47], the in vitro antibacterial activity of electrospun PCL/gelatin/Lawsone (2-hydroxy-1,4-naphthoquinone) nanofiber scaffolds was tested using a disc diffusion assay. Lawsone is the active ingredient of *Lawsonia inermis* L. (commonly known as Henna). The findings indicated that the antibacterial activity of the mats increased with the rising concentration of lawsone. Specifically, PCL/GEL/lawsone 10% exhibited inhibition of the growth of *S. aureus*, aligning with the outcomes observed in the present study.

In future investigations, it is recommended for the kinetic release of the antibacterial additives in a proper solvent to be evaluated. Moreover, a more advanced study can be furthered to investigate the functional properties of the produced fibers.

## 4. Conclusions

In summary, this study explored the potential of 32 endophytic fungi isolated from 13 distinct indigenous Australian plants to inhibit the growth of *S. aureus* and MRSA as well as their capacity to inhibit biofilm formation. Initial antibacterial testing using a disc diffusion assay against *S. aureus* ATCC25923 for both aqueous and EtOAc extracts confirmed the activity of EtOAc extracts of EL 19, EMSF 1-2, EMSF 2-2, and EK 3 against the selected bacteria. Further quantitative evaluation through MIC tests identified EtOAc extracts of EL 19 as the most active sample with an MIC level of 78.1 µg/mL, which remained consistent against antimicrobial-resistant MRSA strains. DNA extraction revealed the active samples as *Chaetomium globosum* (EL 19), *Alternaria* sp. (EMSF 1-2 and EMSF 2-2), and *Coniochaeta* sp. (EK 3), with an interestingly higher E-value for EK 3 (3 × 10^−50^), demonstrating lower confidence in the identification of the isolate below the genus level.

Continuing with EL 19, subsequent testing recorded an MBC level of 156 µg/mL for *S. aureus* ATCC25923 and 625 µg/mL against MRSA M180920. The EL 19 EtOAc extract exhibited significant antibiofilm activity, reducing biofilm formation on polystyrene and preventing cell attachment, with the most substantial reduction observed in *S. aureus* ATCC25923 at $\frac{1}{2}$ × MIC (55% reduction in the CV staining method and 65% reduction in cell viability in the MTT staining method).

Assessing the cytotoxicity of extracts from endophytic fungi incorporated into polymer fibers is crucial for defining the therapeutic window of the designed wound dressing. This step is critically important in functional studies of the wound dressing. Using the BSLT method, the cytotoxicity was demonstrated to be low at up to 6 × MIC (468 µg/mL), with an LC50 value of 639.1 µg/mL. In the final step, the chosen extract was incorporated as an antibacterial additive into the PCL electrospun fibers, and in vitro testing using a disc diffusion assay against *S. aureus* ATCC25923 indicated an increasing inhibition zone with higher concentrations of additives in the produced fiber mats. This suggests the potential of the extracts of endophytes to be employed in electrospun fibers for wound dressing applications. However, for future directions, it would be useful to identify the active secondary metabolites in each endophytic fungi to achieve a better understanding of the metabolites responsible for the antibacterial activity.

## Figures and Tables

**Figure 1 microorganisms-12-01710-f001:**
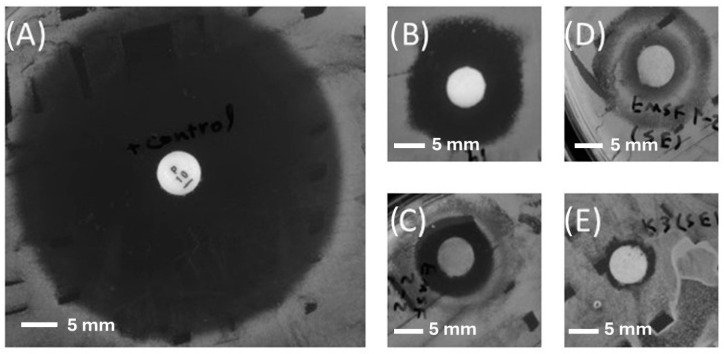
The disc diffusion assay results for (**A**) the positive control (Penicillin 10 µg), (**B**) EL 19 (EtOAc-extracted), (**C**) EMSF 2-2 (EtOAc-extracted), (**D**) EMSF 1-2 (EtOAc-extracted), and (**E**) EK3 (EtOAc-extracted) against *S. aureus* ATCC 25923.

**Figure 2 microorganisms-12-01710-f002:**
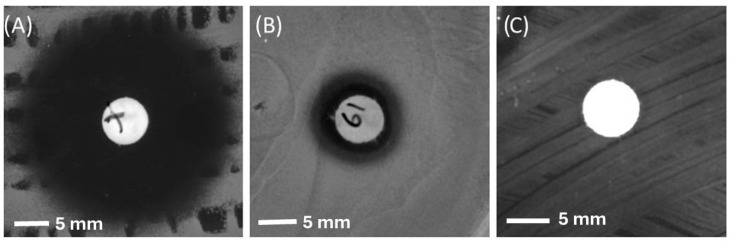
The disc diffusion assay results for (**A**) the positive control (Chloramphenicol 10 µg), (**B**) EL 19 (EtOAc-extracted), and (**C**) Ampicillin (10 µg) against MRSA M180920.

**Figure 3 microorganisms-12-01710-f003:**
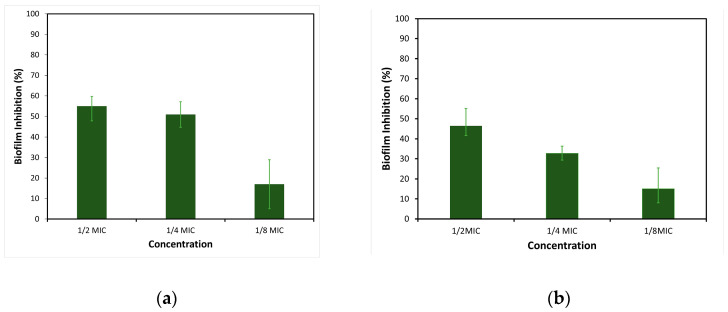
Inhibition of cell attachment in (**a**) *S. aureus* ATCC25923 and (**b**) MRSA M180920 bacterial strains induced by various concentrations of EL 19 EtOAc extracts.

**Figure 4 microorganisms-12-01710-f004:**
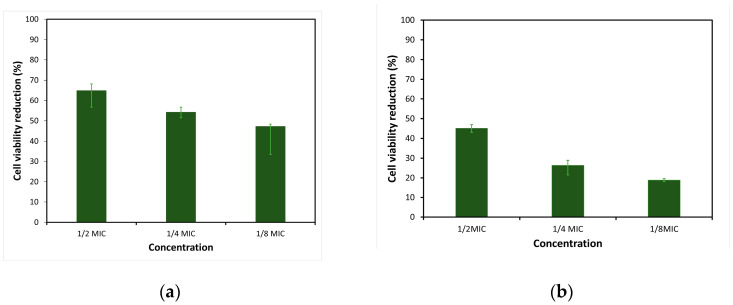
Reduction in cell viability in (**a**) *S. aureus* ATCC25923 and (**b**) MRSA M180920 bacterial strains induced by various concentrations of EL 19 EtOAc extracts.

**Figure 5 microorganisms-12-01710-f005:**
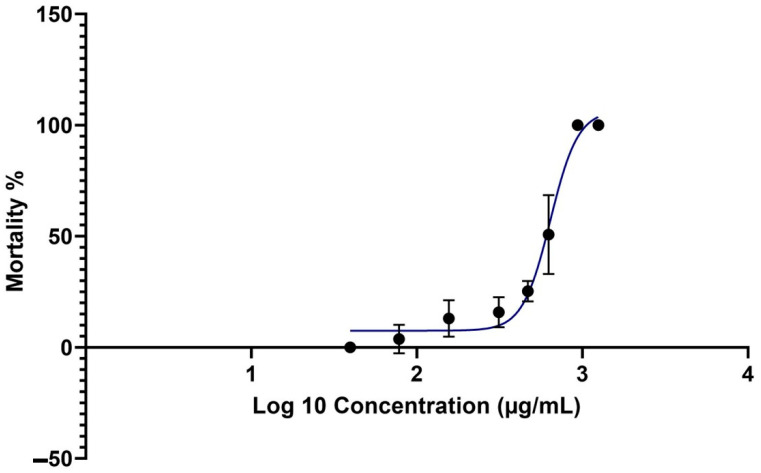
The mortality rate (%) of the brine shrimp nauplii against Log_10_ (Concentration) of EtOAc extracts of EL 19.

**Figure 6 microorganisms-12-01710-f006:**
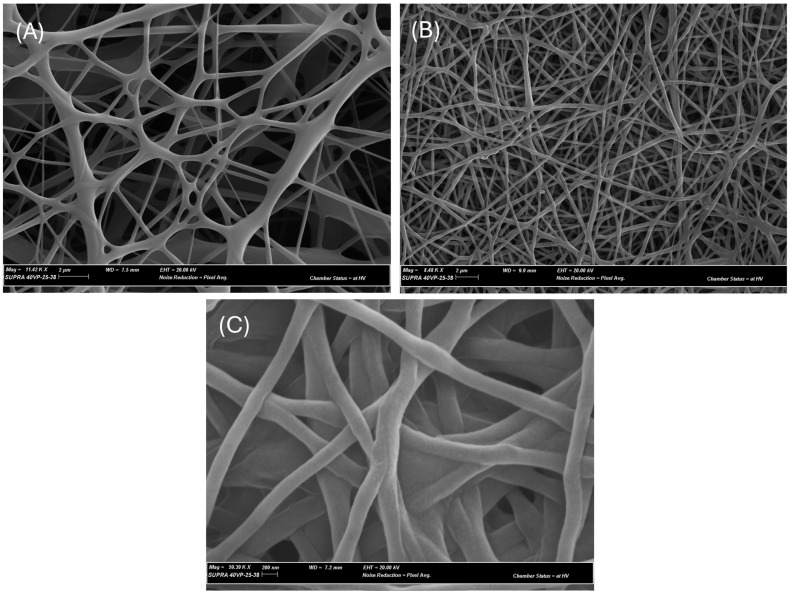
The morphology of 15 wt% PCL fibers (**A**) with no additives, (**B**) 1.89 mg/mL EtOAc extracts of EL 19, and (**C**) 3.716 mg/mL EtOAc extracts of EL 19.

**Figure 7 microorganisms-12-01710-f007:**
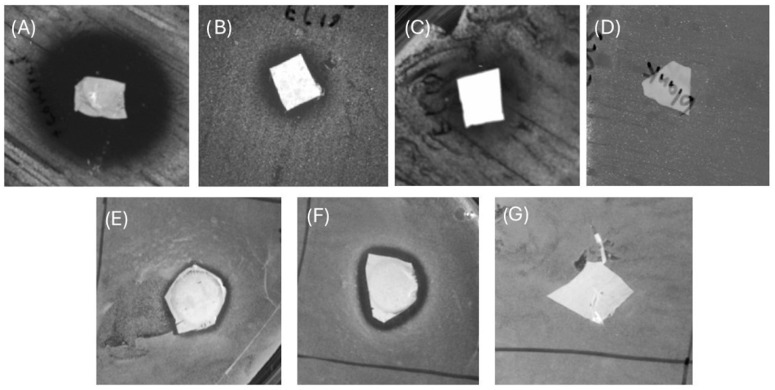
The disc diffusion assay results for 1 mg fiber mats of (**A**) tetracycline in PCL (additive concentration: 53.8 mg/mL), (**B**) EtOAc extracts of EL 19 in PCL (additive concentration: 3.716 mg/mL), (**C**) EtOAc extracts of EL 19 in PCL (additive concentration: 1.89 mg/mL), and (**D**) PCL fibers with no additives against *S. aureus* ATCC25923 and (**E**) EtOAc extracts of EL 19 in PCL (additive concentration: 1.89 mg/mL), (**F**) EtOAc extracts of EL 19 in PCL (additive concentration: 3.716 mg/mL), and (**G**) PCL fibers with no additives against MRSA M180920.

**Table 1 microorganisms-12-01710-t001:** Average zone of inhibition in the screening test against *S. aureus* ATCC 25923.

Sample	Average Zone of Inhibition (mm)
Positive control (Penicillin 10 µg)	39.6
EL 19 (EtOAc-extracted) (1 mg)	11
EMSF 2-2 (EtOAc-extracted) (1 mg)	7.2
EMSF1-2 (EtOAc-extracted) (1 mg)	4
EK3 (EtOAc-extracted) (1 mg)	2
EL 19 (Aqueous-extracted) (1 mg)	0
EMSF 2-2 (Aqueous-extracted) (1 mg)	0
EMSF1-2 (Aqueous-extracted) (1 mg)	0
Negative control (DMSO) (10 µL)	0

**Table 2 microorganisms-12-01710-t002:** Average zone of inhibition in the screening test against MRSA M180920.

Sample	Average Zone of Inhibition (mm)
Positive control (Chloramphenicol 10 µg)	22.1
EL 19 (EtOAc-extracted) (1 mg)	6.5
EL 19 (Aqueous-extracted) (1 mg)	0
Ampicillin 10 µg	0
Negative control (DMSO) (10 µL)	0

**Table 3 microorganisms-12-01710-t003:** MIC results of the active samples against *S. aureus* ATCC25923.

Sample	MIC (µg/mL)
EK 3 (EtOAc-extracted)	>10,000
EMSF 1-2 (EtOAc-extracted)	400
EMSF 2-2 (EtOAc-extracted)	200
EL 19 (EtOAc-extracted)	78.1
Positive control (penicillin)	4.8
Negative control (DMSO)	>10,000

**Table 4 microorganisms-12-01710-t004:** MIC results of EtOAc-extracted compounds obtained from EL 19 against MRSA M180920.

Sample	MIC (µg/mL)
Positive control (Chloramphenicol)	3.6
EL 19 (EtOAc-extracted)	78.1
Penicillin	>10,000
Negative control (DMSO)	>10,000

**Table 5 microorganisms-12-01710-t005:** MBC results of EtOAc extracts obtained from EL 19 against MRSA M180920.

Sample	MBC (µg/mL)	
	*S. aureus* ATCC25923	MRSA 180920
EL19 (EtOAc-extracted)	156	625
Chloramphenicol	12.5	12.5

**Table 6 microorganisms-12-01710-t006:** Identification of the endophytic fungi isolated from *Eremophila longifolia* (EL), *Eremophila maculata* salmon form (EMSF), and *Eremophila kalgoorlie* (EK).

Isolate	E-Value	Species	GenBank Accession Number	Identities (Percentage)
EL 19	0	*Chaetomium globosum*	NR_144860.1	99.81%
EMSF 1-2	2 × 10^−173^	*Alternaria* sp.	KT264731.1	93.93%
EMSF 2-2	2 × 10^−172^	*Alternaria* sp.	KC867287.1	94.75%
EK 3	3 × 10^−50^	*Coniochaeta* sp.	MK163876.1	91.08%

**Table 7 microorganisms-12-01710-t007:** The range of the zone of inhibition of PCL fiber mats against *S. aureus* ATCC25923 and MRSA M180920.

Fiber Sample (PCL 15% *w*/*v*)	Zone of Inhibition (mm)	
	*S. aureus* ATCC25923	MRSA M180920
Positive control/PCL (Tetracycline, C = 53.8 mg/mL)	5.4 ± 0.05	5.3 ± 0.1
EL19 (EtOAc-extracted)/PCL (C = 3.716 mg/mL)	1.4 ± 0.1	1.45 ± 0.05
EL19 (EtOAc-extracted)/PCL (C = 1.89 mg/mL)	0.95 ± 0	0.6 ± 0.05
PCL with no additive	0	0

## Data Availability

Data are available upon reasonable request to the corresponding authors.

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
