# Peer review of "Antibacterial and Antibiofilm Properties of Native Australian Plant Endophytes against Wound-Infecting Bacteria"

_microorganisms, 2024, doi:10.3390/microorganisms12081710_

Round 1
Reviewer 1 Report (New Reviewer)
Comments and Suggestions for Authors
The manuscript Evaluation of Antibacterial and Antibiofilm Properties in Australian Native Plant Endophyte Metabolites Against Wound-infecting Bacteria describes the anti-Staphylococcus aureus activity of the ethyl acetate extract of the endophytic fungus Chaetomium globosum, plus the cytotoxicity assessment of that extract on Artemia salina assay.
Despite the undeniable importance of endophytic fungi as sources of bioactive metabolites with antibacterial activity, the manuscript by Firoozbahr and colleagues is not sufficiently robust. Below, I will outline the main issues that, in my opinion, need to be addressed before sharing the results.
In the Introduction the authors say:
' Out of the 48 endophyte isolates examined, 19 tested positive for the presence of polyketide synthase or non-ribosomal peptide synthetase, as identified through PCR using degenerate primers.' These results were missed.
'The objective of this study was to investigate 32 endophytic fungi, previously isolated by our group from the leaves of 13 distinct species of Australian native plants, for their antibacterial and antibiofilm activities against S. aureus and MRSA species.' No bibliographic references were included.
2. Materials and Methods:
2.1. Preparation of Fungal Extracts from Australian Native Plants:
Delete the plant species with endophytic fungi that were not included.
Extract preparation: light? rotation? rounds of EtOAc extraction? volume extracted/volume EtOAc?
2.4. Determination of MIC:
add the range of final concentrations tested
Define MIC
2.5. Determination of MBC:
Which dilution was applied to apply the samples?
Indicate the cut-off of bacteria considered as the inoculum added.
2.6. Extraction of Fungal DNA:
include in the manuscript the links of sequences uploaded to a free database, GenBank for example.
2.10. Fiber Antibacterial Activity:
Add the mass added to each fiber section tested
Indicate the controls employed.
Indicate how the sterility of the PCL was tested.
3. Results and Discussion:
The antibiotics used as positive controls in the diffusion methods for the beta-lactam-sensitive and beta-lactam-resistant strains were different. The authors should have selected an antibiotic effective against both bacteria, such as vancomycin, for example.
Combine Tables 1 and 2 into a single table. Add the inhibiting activity of extracts inactive in ATCC 25923 on MRSA S. aureus strain. Due to the intrinsic complexity of an extract, and considering that the S. aureus strains differ in their sensitivity to beta-lactams, it is desirable to evaluate all extracts against both strains, as there may be compounds that act through different mechanisms.
Lyophilized extracts of broth cultures concentrate the polar compounds secreted to the medium by the fungi tested and the metabolites extracted with EtOAc. Authors should revise and correct the paragraphs in lines 373-387.
State the values in % of identity considered to cut off for species and genus molecular identification (Table 6).
3.7. PCL Fiber Mat Antibacterial Activity:
When the diffusion assay was carried out, the concentration assayed was 1 mg/ml. Why the concentration of EL19 were major?
Include this result in the manuscript.
Why tetracycline? The positive control should be conserved amongst diffusion assays.
Include the solvent used to dissolve the extract in each assay.
Add a detailed explanation about the determination of the inhibition zones of PCL, replicates, means, and standard deviations.
Why the PCL fiber mat antibacterial activity was tested only on the sensitive strain? The authors should discuss that.
Comments on the Quality of English LanguageMinor issues.
Author Response
Dear Reviewer,
I hope that you are doing well. Please see the attachment.
Best Regards,

Reviewer 2 Report (New Reviewer)
Comments and Suggestions for Authors
The authors did not analyze the composition of the extract for the presence of individual compounds of secondary metabolites. It is known that Chaetomium globosum endophytic fungi synthesize the secondary metabolites with antibacterial activity. Add this to the results discussion (L442-L443). It is appropriate to add the following link to this -
Dwibedi V, Rath SK, Jain S, Martínez-Argueta N, Prakash R, Saxena S, Rios-Solis L. Key insights into secondary metabolites from various Chaetomium species. Appl Microbiol Biotechnol. 2023 Feb;107(4):1077-1093. doi: 10.1007/s00253-023-12365-y.
In conclusion, it is necessary to indicate further prospects for the study, which will consist in the identification of secondary metabolites.
L158 - PDB – decipher
What statistics are shown in Figures 3 and 4? Specify in methods.
The abstract should be a total of about 200 words maximum.
Remove double numbering in Reference.
Author Response
Dear Reviewer,
I hope that you are doing well. Please see the attachment.
Best Regards

Reviewer 3 Report (New Reviewer)
Comments and Suggestions for Authors
Article: microorganisms-3104151-peer-review-v1
Evaluation of Antibacterial and Antibiofilm Properties in Australian Native Plant Endophyte Metabolites Against Wound-infecting Bacteria
The paper should be reviewed and improved considering some of the suggestions mentioned below:
1- The title should be improved and shortened.
2- In affiliation 1, the E-mails that appear in the correspondence should be removed
3- The abstract should be more specific and shorter. Relevant antimicrobial activity values ​​should be included.
4- 2.8. Cytotoxicity
This renowned essay should be reviewed in depth. As presented it is very preliminary.
The presentation of the data is not appropriate. The method used to obtain the lethal concentration or lethal dose 50 (LC or DL) is not mentioned.
For correct presentation of the results, please review the citation mentioned below or another appropriate one.
Us brine shrimp not artemia salina
Statistical analysis
The median lethal concentration (LC50) and 95% confidence
intervals of the test samples were calculated using the probit
analysis method described by Finney[53], as the measure of
toxicity of the plant extract.
Ullah, M. O., Haque, M., Urmi, K. F., Zulfiker, A. H. M., Anita, E. S., Begum, M., & Hamid, K. (2013). Anti–bacterial activity and brine shrimp lethality bioassay of methanolic extracts of fourteen different edible vegetables from Bangladesh. Asian Pacific journal of tropical biomedicine, 3(1), 1-7.
5- 2.9. Fiber Production
This section as presented is very preliminary.
Complete characterization of PCL fibers should be included
6- 3.1. Antibacterial Activity Screening by Disc Diffusion Assay
7- The exact MIC and MBC values ​​for Chloramphenicol and Penicillim must be determined and included in the respective tables.
8- 3.6. Brine Shrimp Lethality test Figure 5 should be replaced by the correctly expressed LC values.
9- The discussion in general is limited to comparison with other extracts and the same strains, example Alternaria. This should be improved, it is not understood what the relevance of the paper is, if there are already studies of other extracts and the same strains that give similar results. The word metabolites is mentioned, however there are no identified compounds. The most active extract has not been analyzed or chemically characterized.
10- The fiber test is carried out at concentrations 2 or 3 times higher than the determined MIC, and the results are not surprising or promising, only halos of moderate inhibitions are reflected. It is necessary to justify the little activity shown.
11- Conclusion
This suggests a strong potential for the secondary metabolites of endophytes to be employed in electrospun fibers for wound dressing applications.
The moderate results at high concentrations shown in Figure 6 do not support the previous sentence.
As the paper is presented, it is not appropriate for the Journal
Comments on the Quality of English LanguageMinor editing of English language are required
Author Response
Dear Reviewer,
I hope that you are doing well. Please see the attachment.
Best Regards

Round 2
Reviewer 3 Report (New Reviewer)
Comments and Suggestions for Authors
The authors have considered the comments and included them in this revised version, which could be considered for acceptance in its current state.
Comments on the Quality of English LanguageMinor editing of English language required.
This manuscript is a resubmission of an earlier submission. The following is a list of the peer review reports and author responses from that submission.
Round 1
Reviewer 1 Report
Comments and Suggestions for Authors
Authors studied the antimicrobial and antibacterial properties of native palnt endophyte metabolites, alone and included in electrospun fiber, against two S. aureus strains.
the paper is interesting and could be useful for the scientific comunity.
in my opinion, the title is uncorrect: to test the fungal extracts against two resistant S. aureus strains (one ATCC and M180920) it doesnt mean to obtain results to Wound care applications.
Please, could authors provide details regarding the toxicity ?
Materials and methods are well described
why the authors didnt perform the CFU determination of the biofilm after treatments? it could be very interesting
the figure 1D is very strange, could author comment more?
Did the authors perform the kinetic of release from fiber, in PBS or Saline solution?
Comments on the Quality of English Language
good
Author Response
Dear Reviewer,
Please find the attachment.
Regards,
Meysam Firoozbahr

Reviewer 2 Report
Comments and Suggestions for Authors
The manuscript authored by Firoozbahr and colleagues is acceptable as a preliminary study. However, as a study that presents robust data, it still needs improvement.
For instance:
The difficulty of publishing preliminary data with plant extracts is the lack of knowledge about the chemical nature of the bioactive compounds.
Suggestions:
In Table 1, data on yield seems lacking; the mass of EtOAc-extracted compounds embedded in the filter to perform the disc inhibitory test.
Line 328, In the paragraph: "As it can be observed from the test results, the EtOAc-extracted metabolites was proved to be more effective in comparison to aqueous crude extracted metabolites suggesting that the active compounds produced by the endophytic fungi are highly polar in nature, which makes them less soluble in the organic solvent"
I couldn't find this observation in the table, figure, or text. Where is the comparison data?
Line 394: Is that correct? "the precision of DNA quantification."
Line 400: what is GCP? What does it mean?
Question: compared with the positive controls (current use antibiotics) the performance of the EtoAc-extracted compounds was inferior. Thus, how could the "hidden compounds" appear as promising effective antimicrobials?
Finally, "wound-healing application" is the end purpose. However, this was not tested. Except for the EtOAc-extracted compounds that were embedded into PCL membranes for assay, the principle of the method was similar to the disc diffusion assay.
It is recommended to proceed with an in vivo or assay as a proof-of-concept or to search for actual wound-healing compounds in Australian plants.
Thus, correcting the manuscript's title is also advisable.
Comments on the Quality of English LanguageMinor editing of English language required
Author Response
Dear Reviewer,
Please see the attachment.
Regards,
Meysam Firoozbahr

Round 2
Reviewer 2 Report
Comments and Suggestions for Authors
The authors responded satisfactorily to most of the reviewer's concerns. So, the manuscript is acceptable in the current revised version.
Author Response
Dear Academic Editor,
Please see the attachment.
Regards,
Meysam Firoozbahr on behalf of Prof. Enzo A. Palombo, Prof. Peter Kingshott, and Dr. Bita Zaferanloo
